# Fruit and Vegetable Consumption and Their Polyphenol Content Are Inversely Associated with Sleep Duration: Prospective Associations from the UK Women’s Cohort Study

**DOI:** 10.3390/nu10111803

**Published:** 2018-11-20

**Authors:** Essra Noorwali, Laura Hardie, Janet Cade

**Affiliations:** 1Nutrition Epidemiology Group, School of Food Science and Nutrition, University of Leeds, Leeds LS2 9JT, UK; j.e.cade@leeds.ac.uk; 2Department of Clinical Nutrition, Faculty of Applied Medical Sciences, Umm Al-Qura University, Makkah 21421, Saudi Arabia; 3Division of Clinical and Population Sciences, Leeds Institute of Cardiovascular and Metabolic Medicine, School of Medicine, University of Leeds, Leeds LS2 9JT, UK; l.j.hardie@leeds.ac.uk

**Keywords:** sleep, fruits and vegetables, polyphenols

## Abstract

This study aims to investigate the prospective associations between fruit and vegetable (FV) intakes and their polyphenol content with subsequent sleep duration in UK women. In this study, 13,958 women with ~4 years of follow-up in the UK Women’s Cohort Study were included in the analyses. FV intakes were assessed at baseline using a food frequency questionnaire (FFQ), and average hours of sleep per day were self-reported in follow-up. Polyphenol intake was calculated by matching FV items from the FFQ with the Phenol-Explorer database. Linear regression models, adjusting for confounders, were used for the analyses. Consuming an additional portion of apples, kiwi, oranges, pineapple, and 100% pure juice were associated with shorter sleep. Similarly, an additional portion of cabbage, celery, aubergine, olives, and peppers were inversely associated with sleep duration. An additional gram of total polyphenols was associated with shorter sleep by 18 min (99% CI −31 to −4, *p* < 0.001). FV consumption and total polyphenol content were inversely associated with sleep duration; however, effect sizes were small, and polyphenol classes from FV intakes were not associated with sleep duration. Future intervention studies considering the time of FV consumption in relation to sleep are needed to clarify the underlying mechanisms.

## 1. Introduction

Epidemiological studies have shown that short sleep duration is associated with hypertension [1], type 2 diabetes [2], cardiovascular disease [3], all-cause mortality [4,5], and a 45% increased risk of obesity compared to normal sleep duration [6,7]. These associations, in part, may be mediated through changes in dietary intake including fruit and vegetables (FV) that influence bodyweight and chronic disease risk [8,9]. However, the directionality of these associations are not well established; a bi-directional relationship has been suggested [10,11]. Recently, St-onge et al. suggested that plant-based diets could have potential benefits for reducing cardiovascular risk by improving sleep [12]. These benefits may be facilitated by dietary polyphenols that have been shown to modulate the circadian rhythms [13] and sleep-wake cycles [14] in rodents. In humans, consuming 2 kiwi fruits an hour before bedtime improved sleep onset, duration, and efficiency in 24 healthy adults during a 4-week open clinical trial [15]. Furthermore, a double-blinded pilot study showed that fresh tart cherry juice reduced insomnia in 15 elderly subjects [16]. The effects of cherries were also observed to increase sleep duration and reduce the number of awakenings, as measured by actigraphy in 12 Spanish participants [17]. These clinical evidence studies suggest sleep-promoting effects of certain fruits; however, they were conducted on small study groups. The limited number of longitudinal epidemiological studies in this area suggests the need for cohort studies with validated dietary intake measures to clarify this association. Therefore, the aim of this study was to explore the prospective associations between specific FV intakes and polyphenol content of FV and sleep duration in a large cohort of UK women. To our knowledge, we are the first to report on prospective associations between specific FV intakes and their polyphenol content with sleep duration using a large cohort. We hypothesized that FV intakes and their polyphenol contents are associated with longer sleep durations.

## 2. Materials and Methods

### 2.1. Participants

The UK Women’s Cohort study (UKWCS) is a large prospective cohort that was established to explore links between diet and chronic diseases. Participants were taken from responders to the World Cancer Research Fund’s direct mail survey including those living in England, Wales, Scotland, and Northern Ireland. Ethical approval was granted at its initiation in 1993 (Research Ethics Committee reference number is 15/YH/0027). The National Research Ethics Committee for Yorkshire and the Humber, Leeds East has now taken on responsibility for the ongoing cohort. [18]. Baseline data collection was between 1995 and 1998 (Figure 1) using a postal food frequency questionnaire (FFQ). Follow-up data (Phase 2) was collected between 1999 and 2002 around 4 years later, and 14,172 women (40% of baseline) completed a follow-up health and life style questionnaire including sleep, and 12,453 women also completed a 4-day food diary and a 1-day activity diary.

### 2.2. Baseline Characteristics

Age, height, weight, medical history, and smoking habits were self-reported. Physical activity was recorded using a binary question in the FFQ which questioned if participants spent time on activities vigorous enough to cause sweating or a faster heartbeat, which indicated moderate physical activity (MET-hours/day). Supplement usage was identified by asking whether participants took any vitamins, minerals, fish oils, fibre, or other food supplements. Participants also self-reported their status regarding vegetarian and vegan diets. Classification of socio-economic status (SES) was undertaken based on occupation, according to the United Kingdom National Statistics-Socio-Economic Classification (NS-SEC), where women are divided into three categories (managerial/professional, intermediate, or routine/manual) [19]. Socio-demographic information such as marital status was determined by self-report questions asking for marital status (married or living as married, divorced, single, widowed, or separated).

### 2.3. Fruit and Vegetable Intakes

Diet was assessed at baseline using a detailed 217-item FFQ developed from the European Prospective Investigation into Cancer and Nutrition (EPIC)-Oxford cohort [20]. The FFQ was validated on a subsample of 303 cohort subjects against a 4-day food diary, as well as fasting blood measures of specific nutrients [18,21,22]. The FFQ were sent to 61,000 participants who had previously responded to a direct mail survey from the World Cancer Research Fund, and a total of 35,692 women completed the FFQ. Participants were asked to choose their frequency of consumption for each food listed in the FFQ by answering the question “how often have you eaten these foods in the last 12 months?” using one of ten response categories; never, less than once a month, 1–3 per month, once a week, 2–4 per week, 5–6 per week, once per day, 2–3 per day, 4–5 per day, and 6+ per day. These were consequently converted to weight of each food consumed per day based on the Food Standards Agency portion sizes book [23]. For the current study, FV food items were used individually, and also their polyphenol contents were calculated from Phenol Explorer [24]. FV consumption was expressed as grams per day (g/day); however, in order to have a better estimate, results were presented per portion size unless stated otherwise, and non-response to FV questions were taken as missing data.

### 2.4. Estimation of Polyphenol Intake from Fruits and Vegetables

The polyphenol intake was calculated by matching FV intakes from the FFQ and the recently developed Phenol Explorer database (www.phenol-explorer.eu) [24] that contains data on the content of 500 polyphenols in over 400 foods. The individual polyphenol intake from each item of FV was calculated by multiplying the content of each polyphenol by the daily consumption of each FV item. Polyphenol classes included flavonoids, phenolic acids, stilbenes, lignans, and other polyphenols (see Appendix B, Table A1, for polyphenol sub-classes). The total polyphenol intake was calculated as the sum of all polyphenol classes intake from all FV reported by the FFQ. This estimation method has been previously used in other studies [25,26,27]. In the Phenol-Explorer database, polyphenol intake was calculated using High performance liquid chromatography (HPLC); however, in the case of lignans and phenolic acids in some FV items (e.g., olives), polyphenol content was calculated using HPLC after hydrolysis, because these treatments are needed to release phenolic compounds that otherwise cannot be analysed [25]. Polyphenol contents from FV items that were analysed using HPLC after hydrolysis are indicated in the tables.

### 2.5. Sleep Duration

Participants were asked about sleep duration using the lifestyle questionnaire in phase 2 in two separate questions in the following form (Appendix A);
“On an average weekday, how is your day spent?”“On an average weekend, how is your day spent?”

Participants were asked to record number of hours and/or minutes that were spent sleeping in an average weekday and weekend. Two separate variables were generated for sleep duration based on weekdays and weekends for all women. Average sleep duration was calculated using the following equation [9]
((minutes slept during the week × 5) + (minutes slept during weekends × 2))/7 (1)

### 2.6. Statistical Analyses

Descriptive statistics, such as means and proportions, described women from the UKWCS according to FV quintiles. We have previously reported from cross-sectional data that sleep duration and FV intakes are non-linearly associated [9]. In order to use linear regression models, we assessed the relationship between FV intakes and sleep duration using locally weighted scatterplot smoothing (LOWESS) [28], which is a tool that creates a smooth line through a scatterplot to help see the trend between variables. The association between total FV intakes (grams/day) and sleep duration (hours/day) showed a linear association (Appendix A). Normal distribution of the outcome variable (sleep duration) was checked using histogram plot (Appendix A). Linear regression models were used to assess the prospective association between FV intakes (continuous exposure in grams/day) or their polyphenols (continuous exposure) and sleep duration (continuous outcome in minutes/day). Unadjusted and adjusted models were used where potential confounders were identified based on previous studies [9]. Confounders were age, SES (professional and managerial, intermediate, routine and manual), smoking (yes, no), ethnicity (white, Bangladeshi, Indian, Chinese, Pakistani, black-Caribbean, black-other, other) and total energy intake. Models that investigated polyphenol intake from FV and sleep duration were further adjusted for other polyphenol content; for example, flavonoids were adjusted for phenolic acids, stilbenes, lignans, and other polyphenols. Total polyphenol intake in association with sleep duration was adjusted for the confounders mentioned above without further adjustment of other polyphenol content. There is not sufficient experimental evidence that body mass index (BMI) and physical activity independently influence fruit and vegetable consumption and their polyphenols to include as an adjustment in the main analyses; however, there is evidence of BMI influencing sleep duration. We adjusted for BMI and physical activity but it did not meaningfully change the results; therefore, the results are shown without adjusting for these covariates. The final number (*n* column in Tables 2–4 and in Appendix B) of women included in the analyses indicates complete data of all covariates included in the model, and difference in (*n*) is due to missing data of any of the included covariates in the model.

Sensitivity analyses were conducted on polyphenol content from FV intakes since they are unstable compounds and can be affected by alcohol intake, supplement use, and medications [29]. Sensitivity analyses included stratifications of certain variables selected prior to analyses between polyphenol content from FV and sleep duration. Variables explored were alcohol consumption frequency (more than once a week, once a week or less, never), BMI (obese vs. non-obese), and dietary habits (vegetarian/vegan vs. non-vegetarian/vegan). Separate analyses were conducted after excluding (1) supplement users, (2) those who self-reported currently having a longstanding illness (see Appendix B
Table A5 for excluded illnesses), and (3) those who self-reported taking prescribed medicines. Further sensitivity analyses included considering weekdays and weekends sleep duration separately, since sleep and dietary habits differ between weekdays and weekends (controlled for confounders stated above in the adjusted model). Further adjustment for BMI and physical activity (MET/hours) in the adjusted model stated above was conducted. To take account of multiple testing, significance level was determined by a *p* value of <0.01 to reflect 99% confidence intervals in main and sensitivity analyses. All analyses were conducted using Stata V. 15.1 statistical software (StataCorp LLC, College Station, TX, USA) [30].

## 3. Results

### 3.1. Socio-Demographic Characteristics

Cohort participants who had unavailable data on sleep duration, extreme sleep duration <2 and >12 h/day, pregnant women, and those with extreme total energy intake <500 and >6000 were excluded, and a total of 13,958 women were included in the analyses (Figure 1). Baseline characteristics of the participants based on FV quintiles are shown in Table 1. The mean age was 52 years (95% CI 52 to 53), and mean BMI was 24.5 (95% CI 24.4 to 24.5). Women had on average 7.5 h/day of sleep (95% CI 7.5 to 7.6) and 24 min/day (95% CI 23 to 25) of physical activity. In total, 32% (95% CI 31 to 32) of women were vegetarian or vegan, 98% (95% CI 98 to 99) were white, and 60% (95% CI 59 to 61) reported supplement usage.

### 3.2. Prospective Associations between Individual FV Items and Sleep Duration

In adjusted models (Table 2), consuming an additional portion of the following fruits were associated with shorter sleep: apples 2 min/day (99% CI −4 to −1, *p* < 0.001), kiwi 10 min/day (99% CI −19 to −2, *p* = 0.001), oranges 3 min/day (99% CI −5 to 0.8, *p* < 0.001), pineapple 17 min/day (99% CI −33 to −1, *p* = 0.006), and 100% pure juice 3 min/day (99% CI −3 to −0.07, *p* = 0.001). Similarly, consuming an additional portion of the following vegetables was associated with shorter sleep: cabbage 6 min/day (99% CI −12 to −0.4, *p* = 0.006), celery 12 min/day (99% CI −23 to −0.5, *p* = 0.007), aubergine 21 min/day (99%CI −41 to −1, *p* = 0.007), olives 37 min/day (99% CI −71 to −3, *p* = 0.004), and peppers 13 min/day (99% CI −23 to −3, *p* = 0.001) (Table 2).

### 3.3. Prospective Associations between Total Polyphenol Content from FV and Sleep Duration

Table 3 shows the prospective associations between total polyphenol intake from FV and sleep duration. In unadjusted analyses (Table 3), an additional gram of flavonoids, lignans, other polyphenols and total polyphenols were inversely associated with sleep duration (*p* < 0.01). In adjusted analyses (Table 3), an additional gram of total polyphenols was associated with shorter sleep by 18 min (99% CI −31 to −4, *p* < 0.001).

### 3.4. Prospective Associations between Polyphenol Classes from FV and Sleep Duration

Table 4 shows the prospective associations between polyphenol classes from FV and sleep duration. In unadjusted analyses (Table 4), flavonoids from apples, kiwi, oranges, cabbage, cucumber, green beans, lettuce, olives, and peppers were negatively associated with sleep duration. Phenolic acids from lettuce, aubergine, olives, and peppers were also negatively associated with sleep duration. Other polyphenols from celery and olives were negatively associated with sleep duration. Lignans from oranges, cabbage, and cucumber were negatively associated with sleep duration.

In adjusted analyses (Table 4), an additional mg of flavonoids from apples was associated with 0.5 min/day shorter sleep (99% CI −0.8 to −0.1, *p* < 0.001), an additional mg of flavonoids from oranges was associated with 0.08 min/day shorter sleep (99% CI −0.1, −0.01, *p* = 0.002), an additional mg of flavonoids from peppers was associated with 3 min/day shorter sleep (99% CI −0.6, −0.4, *p* = 0.003), and an additional mg of phenolic acids from peppers was associated with 31 min/day shorter sleep (99% CI −60 to −2, *p* = 0.005).

### 3.5. Sensitivity Analyses

Sensitivity analyses showed broadly similar results (available Appendix B, Table A2, Table A3, Table A4, Table A5, Table A6 and Table A7). Analyses between polyphenol content from FV and sleep duration stratified by BMI (<25 kg/m^2^ vs. ≥25 kg/m^2^) showed that an additional gram of total polyphenols were associated with shorter sleep by 20 min/day (99% CI −36 to −4, *p* = 0.001) in women with a BMI <25 kg/m^2^, whereas no association was observed in women with a BMI ≥ 25 kg/m^2^ (Appendix B) (Table A2). Stratification of analyses by vegetarian/vegan status showed that an additional gram of total flavonoids were associated with shorter sleep by 52 min/day (99% CI −92 to −13, *p* = 0.001) and total polyphenols by 18 min/day (99% CI −35 to −1, *p* = 0.006) in non-vegan/vegetarian women (Appendix B) (Table A3). When considering weekday and weekend sleep duration separately (Appendix B) (Table A4), total polyphenol intakes were negatively associated with sleep duration on both weekdays and weekends. Analyses between FV intakes and sleep duration stratified by frequency of alcohol consumption showed that total flavonoids from FV were associated with shorter sleep by 44 min/day (99% CI −89 to −0.7, *p* = 0.009) in women consuming alcohol more than once a week (Appendix B) (Table A5). Similarly, total polyphenols from FV were associated with shorter sleep by 23 min/day (99% CI −45 to −2, *p* = 0.005) in women consuming alcohol once a week or less. After excluding women who reported supplement intake, results were attenuated and no association was found between polyphenol intakes and sleep duration (Appendix B) (Table A6), although not significant, polyphenol intakes tended to be negatively associated with sleep duration. After excluding women who reported having a long-term illness, an additional gram of total polyphenols from FV were associated with 18 min/day shorter (99% CI −34 to −3, *p* = 0.001) (Appendix B) (Table A6), and after excluding women who reported medication use, other polyphenols were negatively associated with sleep duration (Appendix B) (Table A6). Results were attenuated between polyphenol intakes and sleep duration after further adjustment of BMI and physical activity (Appendix B) (Table A7); however, the associations tended to be negative.

## 4. Discussion

To our knowledge, this is the first study to report prospective associations between specific FV items and their polyphenol content with sleep duration in a large cohort of UK women. A total of 13,958 women were followed up for approximately 4 years and were included in this study. Our results showed that intakes of apples, kiwi, oranges, pineapple, 100% pure juice, cabbage, celery, cucumber, aubergine, olives, peppers, tomato, and total FV were associated with shorter sleep. In terms of polyphenol content, total polyphenols from FV were negatively associated with sleep duration. Flavonoids from apples, oranges, and peppers were associated with shorter sleep, and phenolic acids from peppers were associated with shorter sleep. Collectively, these findings suggest that among UK women, specific FV items, and total polyphenol intake from FV were associated with shorter sleep. This was not what we had anticipated from previous, smaller scale studies [15,16,17].

Our results are in contrast to some human studies; for example, tart cherry juice has been shown to reduce insomnia severity in older adults in a pilot study [16]. However, no improvement was observed in sleep latency, duration, or efficiency; this may be due to the small sample size (*n* = 15) and the short period of intervention (2 weeks). In addition, the consumption of cherry increased sleep duration and 6-sulfatoxymelatonin (a metabolite that is considered to reflect the nocturnal melatonin concentration) concentrations in the urine in middle aged and elderly participants [17]. However, this study also had a small sample size (*n* = 12) and short period of intervention (3 days). Similarly, the consumption of 2 kiwifruits 1 h before bedtime for 4 weeks increased sleep duration by 13% in adults reporting sleep disturbances [15]. In our results, an additional portion of kiwi intake was associated with 10 min/day shorter sleep (99% CI −19 to −2, *p* = 0.001). Furthermore, in regard to polyphenol intake and sleep, a recent study investigated the effects of chronic supplementation of Holisfiit^®^, a polyphenol-rich extract-based food supplement developed from FV, on both body composition and subjective perception of mental and physical health in 33 overweight and obese participants [31]. After 16-week of supplementation, awakening during the night, total sleep duration, and sleep quality improved by 38% (*p* = 0.04), 50% (*p* = 0.02) and 43% (*p* = 0.03), respectively. In this study, total polyphenol intake was associated with shorter sleep. However, it is important to note that Holisfiit^®^ provided bioactive compounds of polyphenols from flavonoids, delivering flavanones, anthocyanins, and flavanols, and natural components of the methylxanthine family from both an extract of green tea and an extract of yerba mate leaves as well as vitamin B3. Polyphenol effects from supplements differ in bioavailability [32] and concentration to polyphenols from foods [33], which may be one explanation of our different results. Furthermore, a considerable amount of evidence is supporting the hypothesis that high-dose polyphenols from supplements can cause adverse effects through pro-oxidative action, whereas the risk of toxicity from food is low due to poor absorption [33]. Since polyphenols are unstable compounds, some factors need to be considered in comparing our results with previous studies, such as food processing, storage, and ripening stage that impact dietary polyphenol composition [29]. Furthermore, polyphenol content between foods is highly variable, and even within a specific food item the polyphenol content may vary [33]. These conflicting results may be due to the different study designs and participants. Experimental trials on participants with sleep problems differ from healthy free-living individuals; therefore, it is required to consider the potential for non-representative samples taking part in experimental studies.

Our findings are in line with one animal study by Pifferi et al. [14]. They tested the effects of resveratrol dietary supplement, a dietary polyphenolic compound present in FV, on non-human primate grey mouse lemur sleep-wake cycle. After three weeks of resveratrol supplementation, the animals exhibited a significantly-increased proportion of active-wake time occurring during the resting phase (light) of the sleep-wake cycle. Negligible changes in active-wake time during the active phase (dark) of the sleep wake cycle suggesting that resveratrol activity depends largely on the time of administration. Dosing-time dependency of polyphenols were shown previously on melatonin levels [34], the expression pattern of clock genes in the hypothalamus of rats [34], rat tissue lipoperoxidation [35], and adjustment of the clock system in rat liver [36]. Resveratrol administration on male rats behaved as an antioxidant during the night and as a pro-oxidant during day-time [35]. Grape seed proanthocyanidin extract treatment maintained nocturnal melatonin levels and modulated the circadian rhythms when it was administered at the start of the day, rather than at night, in rats [34]. Similarly, grape seed proanthocyanidin modulated the molecular clock by repressing nicotinamide phosphoribosyltransferase (Nampt)—a gene that undergoes transcription by the enhancement of circadian locomotor output cycles (CLOCK)—when the lights were turned on, and overexpressed when the lights were turned off, in rat livers [36]. The effectiveness of polyphenols during periods of the day could be due to the discrepant functionality of the hypothalamus suprachiasmatic nucleus (SCN). It has been shown that SCN cells are extensively coupled during the day, when the cells exhibit synchronous neural activity, but minimally coupled during the night, when the cells are electrically silent [34,37]. Although the timing of FV intakes was not assessed in this study, this could be one explanation of the negative associations we found between FV intakes and sleep duration. FV intake during the day may have a different effect on sleep to when consumed at night. However, as rats are nocturnal animals, in contrast to humans who are diurnal, assessing the zeitgeber time when polyphenol rich foods can entertain circadian rhythms and sleep measures in humans is necessary by conducting interventional trials to determine if there is a time-dependency effect of FV intakes on sleep duration.

Several potential pathways and mechanisms explain the associations between FV intake and their polyphenol content with sleep duration. One of the functions of sleep is to protect the body against the effects of free radicals produced by a high metabolic rate during waking hours [14]. A study suggested that during sleep, metabolic rate and brain temperature are lower than during awake time; this may provide an opportunity to renew the enzymes affected by free radicals. Thus, foods with antioxidant effects are expected to affect the regulation of sleep measures [14]. The action of FV intakes and their polyphenols on sleep measures could be by their improvement on mitochondrial function and energy metabolism by decreasing fat mass, which may lead to changes in sleep [14]. The negative associations between FV intakes and sleep duration in this study could be due to the high antioxidant content in FV that contributes to a decrease in the production of free radicals, and may lead to lower requirements of sleep [14]. It is important to note that the UKWCS contains a higher proportion of vegetarians and well-educated participants, who tend to eat more healthily than the general population; thus, results need to be carefully interpreted. Since the metabolic state of a cell is coupled to the molecular clock, diet may modify rhythmic cellular activities [38]. In light of this, another proposed pathway of how antioxidants may affect sleep is through the protective activation of hormetic, involving proteins such as ion channels, kinases, deacetylases, and transcription factors which regulate the expression of genes that encode cytoprotective enzymes [39]-pathways that promote sirtuin 1(SIRT1) protein expression [40]. SIRT1 has a central role for reactive oxygen species mainly produced as a consequence of mitochondrial functions [41]. It has been identified that several polyphenols, such as resveratrol, act as dietary activators of SIRT1 [40]. In turn, SIRT1 modulates transcription factors including period (PER) 2 [42], which are circadian clock genes that regulate the daily rhythms of locomotor activity, metabolism, and behaviour. Additionally, SIRT1 modulates the ventromedial hypothalamic clock, a brain region that contains neuronal food-synchronized clocks that contribute to regulation of the circadian rhythm in feeding behaviour [43].

Alternatively, polyphenols could adjust the central clock at intestinal levels through the gut-brain axis [34]. Researchers have related the bi-directional interactions between the central nervous system, the enteric nervous system, and the gastrointestinal tract to the prominent role of the gut microbiota in these gut-brain interactions [44]. The bi-directional relationship between microbiota and the circadian system have been shown in germ-free mice that have altered clock gene expression [45,46]. In light of the bi-directional relationship of the gut-brain axis, the polyphenols in FV may affect sleep through nocturnal secretion of melatonin by the pineal gland which is directly controlled in the brain by the SCN. It has been shown that grape seed proanthocyanidin extract administration increased plasma melatonin levels in the middle of the light period, maintaining similar levels at dusk in Male Wistar rats [34]. The beneficial effects of polyphenols may be controlled by the specific microbiota composition of each individual, and there is a strong inter-individual variability in polyphenol bioconversion by the gut microbiota [47]. Some studies suggest that polyphenols have the capacity to alter the gut microbiota composition by increasing the population of beneficial microflora in the gut [48]. It has been previously reported that the bioavailability of polyphenols and the presence of bioactive metabolites in rat plasma depend on the rat sex and the amount ingested [49]. Recently, diverse effects of resveratrol in induced colitis in mice depended on the sex of the animal [50]. Our studies included only women, and further human studies are needed to confirm if the effects of FV intakes and their polyphenols differ by sex.

Although several mechanisms have been proposed on how FV intakes and their polyphenols may affect sleep and the circadian system, more studies are needed to define the mechanisms by which polyphenols could adjust sleep measures and the central clock. Future studies that can demonstrate that the effects mediated by polyphenol supplements mimic the effects of whole foods are essential. Furthermore, dose-response and time of administration-response studies allowing for identification of the most effective doses and times are needed. Studies demonstrating that some polyphenols are transported to the brain or are present in the circulation at the times of the beneficial effects on sleep measures are important to clarify the underlying mechanisms between FV intakes and sleep.

### Strengths and Limitations

In interpreting the results of our analyses, certain limitations of the study should be considered. Dietary intakes were collected at one time point only, which means any changes in dietary pattern over time were not taken into account. Self-report of FV intakes in the UKWCS are above the national average [51], possibly due to over-reporting through the FFQ [52], which was observed in other cohort studies using this dietary assessment method [53]. The method to assess sleep duration has not been validated, and self-reported sleep duration may cause over-reporting [54], and any change in sleep patterns were not taken into account. In addition, it was difficult to exclude the effects of other dietary polyphenols from other sources such as caffeine, red wine, and tea [55]. Whilst inverse associations between FV intakes and sleep duration have been observed, effect size was small, and may not be clinically significant. Interpretation of the extent of causality should be undertaken with caution, since observational studies have substantial potential for biases caused by incomplete adjustments for confounders, measurement error in the exposure estimate, and biases in participation selection. On the other hand, our analyses have several strengths. The UKWCS is a large prospective cohort which includes health-conscious women with wide diversity in dietary intakes, which facilitates clarifying the associations between FV intakes and sleep duration. Furthermore, to our knowledge, this is the first study that has extensively investigated the associations between FV items and their polyphenol content on sleep duration in UK women using a validated dietary assessment tool. The use of Phenol Explorer as a reference database for polyphenol content has several advantages due to the high quality literature articles on polyphenol composition, the impacts of food processing on the polyphenols, and metabolite composition in the body. These advantages ensure that the polyphenol content of FV applied here were sensible with regard to the variety of polyphenols in each FV item.

## 5. Conclusions

FV intakes and their polyphenol content were associated with shorter sleep in a sub-group of UK women. Further investigations are required to assess the relationship between FV intakes and sleep measures in the general population using objective methods of sleep. Overall, the findings of this study do not provide strong evidence to suggest that polyphenol classes from FV intakes are important in relation to sleep duration. Until further knowledge is obtained from intervention studies, consumption of five or more servings/day of FV and sleeping 7–9 h/day for adults is recommended.

## Figures and Tables

**Figure 1 nutrients-10-01803-f001:**
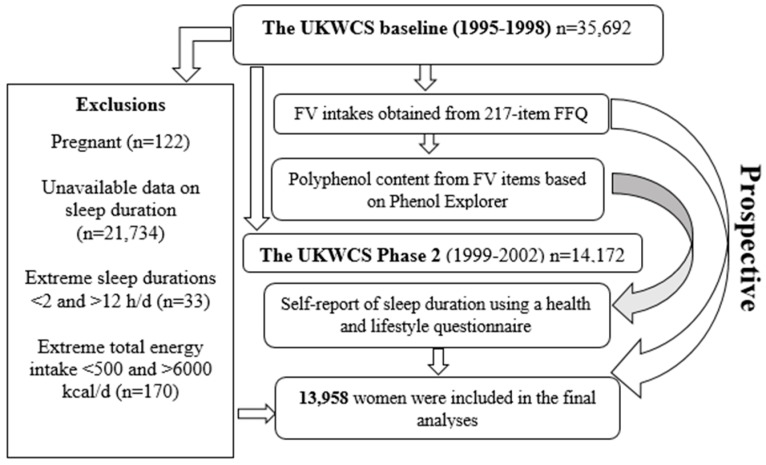
Participants’ flow chart. d (day), FFQ (food frequency questionnaire), FV (fruits and vegetables), h (hours), UKWCS (The UK Women’s Cohort Study).

**Table 1 nutrients-10-01803-t001:** Baseline characteristics by quintile of total FV intakes from the UKWCS.

Characteristics	0–348	348–486	486–624	624–817	817–1597	Total
Number of Women (*n*)	2364	2710	2873	2996	3015	13,958
	Mean (95% CI)	Mean (95% CI)	Mean (95% CI)	Mean (95% CI)	Mean (95% CI)	Mean (95% CI)
Age (years)	51 (50, 51)	52 (51, 52)	52 (52, 52)	52 (52, 52)	53 (52, 53)	52 (52, 53)
BMI (kg/m^2^)	24.4 (24.2, 24.5)	24.1(24.0, 24.3)	24.2 (24.0, 24.3)	24.0 (23.9, 24.2)	24.1 (23.9, 24.2)	24.5(24.4, 24.5)
Total energy intake (kcal/day)	2036 (2004, 2069)	2115 (2092, 2138)	2272 (2249, 2295)	2422 (2398, 2446)	2698 (2672, 2724)	2307 (2299, 2314)
Sleep duration *	7.5 (7.5, 7.6)	7.5 (7.5, 7.6)	7.5 (7.5, 7.5)	7.5 (7.5, 7.6)	7.4 (7.4, 7.5)	7.5 (7.5, 7.6)
Weekday sleep duration *	7.5 (7.4, 7.5)	7.5 (7.5, 7.5)	7.5 (7.4, 7.5)	7.5 (7.4, 7.5)	7.4 (7.4, 7.4)	7.5 (7.4, 7.5)
Weekend sleep duration *	7.9 (7.9, 8.0)	7.9 (7.9, 7.9)	7.9 (7.8, 7.9)	7.9 (7.8, 7.9)	7.7 (7.7, 7.8)	7.9 (7.8, 7.9)
Physical activity (minutes/day)	27 (24, 29)	21 (20, 22)	23 (21, 24)	23 (22, 24)	28 (26, 29)	24 (23, 25)
	% (95% CI)	% (95% CI)	% (95% CI)	% (95% CI)	% (95% CI)	% (95% CI)
Has longstanding illness (yes)	25 (23, 27)	24 (23, 26)	26 (24, 28)	27 (25, 28)	28 (26, 29)	26 (25, 27)
Prescribed medicine (yes)	31.2 (29, 33)	32.1 (30, 33)	30.6 (28, 32)	31.8 (30, 33)	31.8 (30, 33)	31.5 (30, 32)
Smoking (yes)	13 (12, 14)	9 (8, 10)	8 (7, 9)	6 (5, 7)	6 (5, 7)	8 (8, 9)
Supplement use (yes)	55 (53, 57)	56 (54, 57)	59 (56, 60)	62 (60, 63)	67 (65, 69)	60 (59, 61)
Vegetarian or vegan (yes)	24 (22, 26)	27 (25, 28)	30 (28, 31)	34 (33, 36)	42 (40, 43)	32 (31, 32)
Ethnicity (white)	98 (98, 99)	99 (98, 99)	99 (98, 99)	98 (98, 99)	98 (97, 98)	98 (98, 99)
Employer (employed)	62 (60, 63)	60 (58, 62)	59 (57, 61)	58 (56, 60)	56 (54, 57)	59 (58, 60)
SES (professional)	61 (59, 63)	63 (61, 65)	67 (65, 69)	67 (65, 68)	70 (68, 71)	66 (65, 66)
Marital status (married)	73 (71, 75)	75 (74, 73)	77 (75, 78)	77(75, 78)	77 (75, 78)	76 (75, 77)
Number of children (2 children)	51 (49, 54)	52 (50, 54)	51 (49, 53)	48 (46, 51)	49 (47, 51)	50 (49, 51)

* Hours/day, BMI (body mass index), CI (confidence interval), FV (fruit and vegetables), mg (milligram), µg (microgram), m (metre), SES (socio-economic status), UKWCS (The UK Women’s Cohort Study).

**Table 2 nutrients-10-01803-t002:** The prospective associations between FV intakes and sleep duration of women from the UKWCS.

Sleep Duration (minutes/day)						
Models	Unadjusted			Adjusted *		
**Fruit**	**Coefficient per Additional 80 g/day (99% CI)**	***p* Value**	***n***	**Coefficient per Additional 80 g/day (99% CI)**	***p* Value**	***n***
Apples	−2 (−4, −1)	<0.001	13,530	−2 (−4, −1)	<0.001	12,862
Avocados	−8 (−22, 4)	0.1	8468	−8 (−23, 5)	0.1	8033
Bananas	−0.4 (−2, 1)	0.5	13,092	−0.6 (−2, 1)	0.4	12,437
Grapes	−1 (−4, 0.5)	0.05	13,312	−1 (−3, 1)	0.2	12,664
Kiwi	−10 (−19, −2)	0.001	11,709	−10 (−19, −2)	0.001	11,147
Mangoes	−6 (−16, 4)	0.1	7293	−6 (−17, 4)	0.1	6930
Oranges, satsumas, grapefruit	−2 (−4, 0.4)	0.002	12,967	−3 (−5, −0.8)	<0.001	12,330
Papaya	−4 (−18, 9)	0.4	3646	−5 (−20, 9)	0.3	3445
Pears	−0.3 (−3, 2)	0.7	12,177	−0.1 (−3, 2)	0.8	11,576
Pineapple	−20 (−35, −4)	0.001	11,810	−17 (−33, −1)	0.006	11,243
Apricots	−44 (−89, 1)	0.01	11,010	−40 (−88, 6)	0.02	10,456
Melon	−1 (−4, 0.8)	0.07	13,110	−1 (−4, 1)	0.2	12,478
Nectarines	−3 (−26, 19)	0.6	12,678	−8 (−32, 14)	0.3	12,062
Peaches	−32 (−71, 6)	0.03	12,945	−32 (−73, 8)	0.04	12,315
Plums	−29 (−74, 16)	0.09	12,839	−31 (−77, 15)	0.08	12,217
Raspberries	−4 (−14, 6)	0.2	12,643	−2 (−13, 7)	0.4	12,040
Redcurrants, blackcurrants	−4 (−12, 2)	0.09	10,426	−4 (−11, 3)	0.1	9910
Rhubarb	1 (−2, 4)	0.3	11,359	1 (−2, 4)	0.3	10,829
Strawberries	−12 (−31, 6)	0.09	13,321	−11 (−31, 8)	0.1	12,673
Orange juice-pure fruit per 125 g/day	−1 (−3, 0.5)	0.06	12,224	−1 (−3, 0.4)	0.04	11,642
100% pure fruit juice per 125 g/day	−3 (−6, −0.08)	0.001	10,345	−3 (−6, −0.07)	0.001	9834
**Dried fruit per 25 g/day**						
Dates	−2 (−8, 3)	0.2	9577	−3 (−9, 3)	0.1	9117
Figs	−0.008 (−3, 3)	0.9	7231	−0.2 (−3, 3)	0.8	6860
Prunes	−4 (−11, 2)	0.09	9205	−4 (−11, 2)	0.1	8739
Mixed dried fruit	−0.9 (−6, 4)	0.6	8972	−0.8 (−6, 4)	0.6	8531
Currants, raisins, sultanas	0.4 (−6, 7)	0.8	12,605	0.8 (−6, 8)	0.7	12,003
**Vegetables**						
Beetroot	−16 (−35, 1)	0.01	11,516	−12 (−31, 7)	0.1	10,951
Broccoli, spring greens, kale	−2 (−7, 1)	0.1	13,488	−1 (−6, 3)	0.5	12,822
Brussels	−3 (−9, 2)	0.1	12,783	−0.6 (−7, 6)	0.8	12,153
Cabbage	−8 (−13, −3)	<0.001	13,051	−6 (−12, −0.4)	0.006	12,425
Carrots	−2 (−7, 2)	0.1	13,713	−1 (−6, 3)	0.5	13,037
Cauliflower	−0.04 (−0.1, 0.01)	0.06	13,503	−0.03 (−0.1, 0.03)	0.2	12,842
Celery	−14 (−25, −3)	0.001	12,483	−12 (−23, −0.5)	0.007	11,865
Coleslaw	−6 (−27, 15)	0.4	10,018	−4 (−27, 17)	0.5	9546
Courgettes, marrow, squash	−6 (−14, 2)	0.05	11,697	−7 (−16, 1)	0.03	11,134
Cucumber	−17 (−32, −1)	0.005	12,911	−15 (−32, 0.8)	0.01	12,293
Green and runner beans	−6 (−11, −1)	0.001	13,344	−5 (−10, 0.6)	0.02	12,688
Lettuce	−15 (−31, −0.1)	0.009	13,550	−14 (−30, 2)	0.02	12,885
Aubergine, okra	−25 (−44, −6)	<0.001	8258	−21 (−41, −1)	0.007	7853
Olives	−36 (−69, −3)	0.005	6461	−37 (−71, −3)	0.004	6129
Parsnips	−0.7 (−12, 10)	0.8	11,704	1 (−19, 13))	0.6	11,137
Peas, mushy peas, mange tout	−5 (−15, 3)	0.1	13,132	−5 (−15, 4)	0.1	12,510
Peppers	−10 (−20, −1)	0.004	12,371	−13 (−23, −3)	0.001	11,780
Swedes	−0.07 (−11, 11)	0.9	11,052	1 (−10, 13)	0.7	10,512
Tomatoes-raw, canned, sauce	−2 (−5, 0.1)	0.01	13,526	−2 (−5, 0.1)	0.01	12,872
Turnip	−5 (−20, 9)	0.3	8998	−5 (−21, 10)	0.3	8568
Mustard, cress, watercress	−21 (−73, 31)	0.3	11,283	−21 (−75, 32)	0.3	10,736
Boiled/mashed potatoes	0.3 (−1, 1)	0.5	13,330	0.9 (−0.6, 2)	0.1	12,692

* Adjusted for age, socio-economic status, smoking, ethnicity, and total energy intake.

**Table 3 nutrients-10-01803-t003:** The prospective associations between total polyphenols from FV consumption and sleep duration in the UKWCS.

	Sleep Duration (minutes/day)					
	Unadjusted			Adjusted *		
Polyphenol Class	Coefficient per Additional Gram (99% CI)	*p* Value	*n*	Coefficient per Additional Gram (99% CI)	*p* Value	*n*
Total flavonoids	−30 (−54, −6)	0.001	13,636	−30 (−61, 1)	0.01	12,816
Total phenolic acids	−22 (−61, 17)	0.1	13,635	22 (−30, 74)	0.2	12,816
Total other polyphenols	−180 (−330, −30)	0.002	13,805	−136 (−299, 27)	0.03	12,816
Total stilbenes	−4011 (−8731, 708)	0.03	13,670	−538 (−6418, 5341)	0.08	12,816
Total lignans	−28 (−54, −2)	0.005	13,880	−14 (−43, 15)	0.2	12,816
Total polyphenols from FV **	−16 (−28, −5)	<0.001	13,636	−18 (−31, −4)	0.001	12,971

* Adjusted for age, socio-economic status, smoking, ethnicity, total energy intake, and other polyphenol components. ** Total polyphenols (not adjusted for other polyphenol components) = the sum of total flavonoids, total phenolic acids, total other polyphenols, total stilbenes, and total lignans.

**Table 4 nutrients-10-01803-t004:** The prospective associations between polyphenol classes from FV consumption and sleep duration in the UKWCS.

	Sleep Duration (minutes/day)					
	Unadjusted			Adjusted *		
Polyphenol Classes	Coefficient per Additional mg (99% CI)	*p* Value	*n*	Coefficient per Additional mg (99% CI)	*p* Value	*n*
**Flavonoids**						
Apple	−0.5 (−0.8, −0.2)	<0.001	13,530	−0.5 (−0.8, −0.1)	<0.001	12,536
Avocados	−19 (−50, 11)	0.1	8468	−7 (−41, 26)	0.5	7900
Banana	−0.1 (−0.6, 0.4)	0.5	13,092	−0.1 (−0.7, 0.4)	0.5	12,115
Grape	−0.3 (−0.8, 0.1)	0.05	13,312	−0.001 (−0.5, 0.5)	0.9	12,413
Kiwi	−19 (−34, −5)	0.001	11,709	−15 (−31, 0.8)	0.02	10,939
Mangoes	−4 (−11, 2)	0.1	7293	−1 (−9, 7)	0.6	6831
Oranges **	−0.07 (−0.13, 0.01)	0.002	12,967	−0.08 (−0.1, −0.01)	0.002	12,054
Pears	−0.09 (−0.81, 0.62)	0.7	12,177	0.1 (−0.6, 0.9)	0.5	11,352
Apricots	−7 (−14, 0.3)	0.01	11,010	−4 (−12, 4)	0.2	10,304
Nectarines	−0.3 (−2, 1)	0.6	12,678	−0.3 (−2, 1)	0.6	11,824
Peaches	−17 (−38, 3)	0.03	12,945	−13 (−39, 11)	0.1	12,063
Plums	−0.3 (−0.9, 0.2)	0.09	12,839	−0.3 (−1, 0.2)	0.1	11,981
Raspberries	−0.05 (−0.1, 0.08)	0.2	12,643	−0.03 (−0.2, 0.1)	0.6	11,815
Redcurrants	−0.1 (−0.3, 0.08)	0.09	10,426	−0.02 (−0.6, 0.5)	0.9	9784
Rhubarb	0.4 (−0.81, 1)	0.3	11,359	0.9 (−0.4, 2)	0.09	10,620
Strawberries	−0.1 (−0.4, 0.1)	0.09	13,321	−0.1 (−0.4, 0.2)	0.4	12,433
Orange juice (pure fruit)	−0.02 (−0.06, 0.01)	0.06	12,224	−0.01 (−0.05, 0.02)	0.3	12,054
Figs	−0.2 (−121, 120)	0.9	7231	−6 (−141, 127)	0.8	6739
Prunes	−7 (−17, 3)	0.09	9205	−8 (−21, 4)	0.08	8565
Raisins	3 (−49, 56)	0.8	12,605	6 (−52, 64)	0.7	11,726
Beetroot **	−42 (−87, 3)	0.02	11,516	−27 (−77, 22)	0.1	10,681
Broccoli	−0.1 (−0.3, 0.08)	0.1	13,488	−0.01 (−0.2, 0.2)	0.8	12,490
Brussels **	−3 (−8, 2)	0.1	12,783	0.8 (−5, 7)	0.7	11,846
Cabbage **	−262 (−431, −93)	<0.001	13,051	−180 (−373, 11)	0.02	12,107
Cucumber **	−209 (−403, −15)	0.005	12,911	−165 (−372, 41)	0.04	12,002
Green beans	−1 (−1, −0.2)	0.001	13,344	−0.4 (−1, 0.4)	0.1	12,363
Lettuce	−4 (−9, −0.03)	0.009	13,550	−3 (−8, 1)	0.07	12,555
Olives	−0.2 (−0.5, −0.02)	0.005	6461	1 (−0.8, 2)	0.1	6049
Parsnips **	−0.9 (−15, 13)	0.8	11,705	4 (−11, 19)	0.4	10,885
Peas	−356 (−939, 226)	0.1	13,132	−272 (−908, 364)	0.2	12,195
Peppers	−3 (−5, −0.3)	0.004	12,371	−3 (−6, −0.4)	0.003	11,527
Swedes **	−0.01 (-2, 2)	0.9	11,052	0.4 (−1, 2)	0.6	10,268
Tomatoes	−10 (−22, 0.5)	0.01	13,526	−8 (−20, 4)	0.08	12,534
Watercress **	−1 (−6, 2)	0.3	11,283	−0.6 (−5, 4)	0.7	10,531
**Phenolic acids**						
Bananas	−0.5 (−3, 2)	0.5	13,092	−0.2 (−3, 2)	0.8	12,115
Grapes (green)	−0.2 (−0.4, 0.06)	0.05	13,312	−0.008 (−0.3, 0.3)	0.9	12,413
Pears	−0.03 (−0.3, 0.2)	0.7	12,177	0.1 (−0.1, 0.4)	0.3	11,352
Apricots	−5 (−11, 0.2)	0.01	11,010	−2 (−9, 3)	0.2	10,304
Nectarines	−0.4 (−3, 2)	0.6	12,678	0.2 (−3, 3)	0.8	11,824
Peaches	−1 (−3, 0.3)	0.03	12,945	−0.7 (−3, 1)	0.4	12,063
Plums	−0.4 (−1, 0.2)	0.09	12,839	−0.2 (−0.9, 0.5)	0.4	11,981
Raspberries	−0.04 (−0.1, 0.06)	0.2	12,643	0.03 (−0.1, 0.1)	0.5	11,815
Redcurrants	−2 (−5, 1)	0.09	10,426	−0.2 (−7, 7)	0.9	9784
Strawberries	−1 (−3, 0.6)	0.09	13,321	−0.1 (−2, 2)	0.9	12,433
Dates	−0.4 (−1, 0.5)	0.2	9577	−0.3 (−1, 0.7)	0.4	8942
Prunes	−0.09 (−0.2, 0.04)	0.09	9205	−0.07 (−0.2, 0.07)	0.2	8565
Raisins	0.1 (−1, 2)	0.8	12,605	0.3 (−1, 2)	0.6	11,726
Broccoli	−0.1 (−0.5, 0.1)	0.1	13,488	0.05 (−0.3, 0.4)	0.6	12,490
Carrots	−0.1 (−0.4, 0.1)	0.1	13,713	−0.03 (−0.3, 0.3)	0.7	12,679
Cauliflower	−0.6 (−1, 0.2)	0.06	13,503	−0.2 (−1, 0.8)	0.5	12,509
Lettuce	−5 (−10, −0.03)	0.009	13,550	−3 (−9, 2)	0.1	12,555
Aubergine	−21 (−36, −5)	<0.001	8258	−14 (−31, 2)	0.03	7735
Olives	−0.3 (−0.6, −0.02)	0.005	6461	−0.5 (−3, 2)	0.6	6049
Peppers	−29 (−55, −3)	0.004	12,371	−31 (−60, −2)	0.005	11,527
Tomatoes	−0.8 (−1, 0.04)	0.01	13,526	−0.4 (−1, 0.4)	0.2	12,534
Boiled/mashed potatoes	0.01 (−0.04, 0.07)	0.5	13,330	0.04 (−0.02, 0.1)	0.1	12,346
**Other polyphenols**						
Pears	−9 (−78, 60)	0.7	12,177	28 (−47, 103)	0.3	11,352
Orange juice (pure fruit)	−0.5 (−1, 0.2)	0.06	12,224	1 (−0.02, 2)	0.01	11,418
Celery	−6 (−12, −1)	0.001	12,483	−4 (−10, 0.8)	0.02	11,623
Olives	−0.1 (−0.3, −0.01)	0.005	6461	−0.1 (−0.3, 0.02)	0.03	6049
**Stilbenes**						
Grapes	−7 (−17, 2)	0.05	13,312	−1 (−13, 10)	0.7	12,413
Redcurrants	−3 (−9, 2)	0.09	10,426	−0.1 (−7, 6)	0.9	9784
Strawberries	−44 (−114, 24)	0.09	13,321	−15 (−98, 67)	0.6	12,433
**Lignans**						
Oranges **	−0.2 (−0.4, −0.03)	0.002	12,967	−0.2 (−0.4, 0.05)	0.03	12,054
Pineapple **	−0.1 (−0.2, −0.02)	0.001	11,810	−0.09 (−0.2, 0.004)	0.01	11,027
Melon *	−0.02 (−0.05, 0.009)	0.07	13,110	−0.005 (−0.04, 0.02)	0.6	12,195
Brussels **	−0.06 (−0.1, 0.04)	0.1	12,783	0.006 (−0.1, 0.1)	0.9	11,846
Cabbage **	−132 (−218, −47)	<0.001	13,051	−89 (−185, 6.8)	0.01	12,107
Squash **	−862 (−2011, 287)	0.05	11,697	−874 (−2124, 375)	0.07	10,914
Cucumber **	−5 (−10, −0.4)	0.005	12,911	−4 (−9, 0.9)	0.03	12,002
Parsnips **	−19 (−303, 264)	0.8	11,705	70 (−233, 374)	0.5	10,885
Swedes **	−19 (−2822, 2783)	0.9	11,052	434 (−2671, 3539)	0.7	10,268
Turnip **	−45 (−173, 83)	0.3	8998	−57 (−194, 80)	0.2	8381

* Adjusted for age, socio-economic status, smoking, ethnicity, total energy intake and other polyphenol components. ** Analysed using chromatography after hydrolysis.

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
