# Peer review of "Fruit and Vegetable Consumption and Their Polyphenol Content Are Inversely Associated with Sleep Duration: Prospective Associations from the UK Women’s Cohort Study"

_nutrients, 2018, doi:10.3390/nu10111803_

Round 1
Reviewer 1 Report
The study investigates the possible association between fruit and vegetable intakes on sleep duration.
The paper is linked to UK Women's Cohort study but the referee indicates to add also men in the study for considering the gender in sleep duration.
This can be important for the readers since the scientists have more attention in sex differences in many fields.
The Authors could report also other results regarding oxidative stress since they discuss about this pathway in the discussion
Author Response
Dear editor,
I would like to thank the reviewers for their valuable comments on the manuscript. We have revised the manuscript in light of these comments.
We hope that our revised manuscript has successfully addressed these points and will be considered for publication.
Best wishes,
Essra Noorwali
Reviewer 1 comments
The study investigates the possible association between fruit and vegetable intakes on sleep duration.
The paper is linked to UK Women's Cohort study but the referee indicates to add also men in the study for considering the gender in sleep duration.
We appreciate your comment. Unfortunately, the UK Women’s Cohort study only collected data on women therefore, it is not possible to include men in the study. It would be an interesting study to conduct in another dataset that include women and men to explore sex differences in the association between fruit and vegetable consumption and sleep duration.
This can be important for the readers since the scientists have more attention in sex differences in many fields.
The Authors could report also other results regarding oxidative stress since they discuss about this pathway in the discussion
Thank you for your comment. Oxidative stress is the result of the imbalance between reactive oxygen species (ROS) formation and enzymatic and nonenzymatic antioxidants. There are several biomarkers of oxidative stress that may be useful to assess the effects of antioxidants (since polyphenols contain antioxidants) on sleep. These include 1) ROS in leucocytes and platelets, 2) ROS induced modifications of lipids, DNA and proteins,3) enzymatic players of redox status, 4) total antioxidant capacity of human body fluids (Marrocco et al. 2017). However, The UK Women’s Cohort Study focused on diet and health outcomes. Unfortunately no suitable biological specimens were available to assess oxidative stress in this study.
It would be an interesting study to conduct to understand the underlying mechanisms linking fruit and vegetable consumption and their polyphenol content with sleep measures. We have mentioned in the discussion (lines 341-349) that more studies are needed to define the mechanisms by which polyphenols could adjust sleep measures and the central clock and considering dose-response and time of administration. Unfortunately, reporting results regarding oxidative stress in this was not feasible.
Ilaria Marrocco, Fabio Altieri, and Ilaria Peluso, “Measurement and Clinical Significance of Biomarkers of Oxidative Stress in Humans,” Oxidative Medicine and Cellular Longevity, vol. 2017, Article ID 6501046, 32 pages, 2017. https://doi.org/10.1155/2017/6501046.
Reviewer 2 Report
Noorwali wt al. report on the inverse association between ruit and vegetable consumption and sleep duration using a UK Women’s Cohort Study. The topic is interesting and overall the research well conducted. It would be interesting to relate the shorter sleep parameter the authors report in the results section with the different composition of the tested fruit and vegetables. Also, effect sizes were small, so it is difficult to understand the significance of the study.
Author Response
Dear editor,
I would like to thank the reviewers for their valuable comments on the manuscript. We have revised the manuscript in light of these comments.
We hope that our revised manuscript has successfully addressed these points and will be considered for publication.
Best wishes,
Essra Noorwali
Noorwali et al. report on the inverse association between fruit and vegetable consumption and sleep duration using a UK Women’s Cohort Study. The topic is interesting and overall the research well conducted. It would be interesting to relate the shorter sleep parameter the authors report in the results section with the different composition of the tested fruit and vegetables. Also, effect sizes were small, so it is difficult to understand the significance of the study.
Thank you for your interest and comment. The bidirectional association between sleep and diet has been suggested by other researchers (see lines 37-38). However, this study focused on fruit and vegetable consumption as the exposure and sleep duration as the outcome. We have conducted previous analyses relating short and long sleep durations with fruit and vegetable intakes using a nationally representative sample (see reference 9). Changes in sleep with the different composition of polyphenol classes are stated in table 3 and with specific polyphenols in fruit and vegetable items are stated in table 4.
We are aware of the limitations of the study and that the effect sizes were small (this was mentioned in the abstract, see line 27). This study is a starting point to assess whether an association exists between fruit and vegetable consumption and sleep duration. Few studies have assessed this association and this study may provide insight for future studies to assess the underlying mechanisms that we discussed (see lines 299-340). Sleep and fruit and vegetable consumption are important lifestyle factors that influence the risk of chronic diseases and understanding the relationship between them is vital.